# NAD Supplement Alleviates Intestinal Barrier Injury Induced by Ethanol Via Protecting Epithelial Mitochondrial Function

**DOI:** 10.3390/nu15010174

**Published:** 2022-12-30

**Authors:** Wenli Li, Yujia Zhou, Nengzhi Pang, Qianrong Hu, Qiuyan Li, Yan Sun, Yijie Ding, Yingying Gu, Ying Xiao, Mengqi Gao, Sixi Ma, Jie Pan, Evandro Fei Fang, Zhenfeng Zhang, Lili Yang

**Affiliations:** 1Guangdong Provincial Key Laboratory of Food, Nutrition and Health, Department of Nutrition, School of Public Health, Sun Yat-sen University, No. 74 Zhongshan Road 2, Yuexiu District, Guangzhou 510080, China; 2Department of Immunization Programmes, Guangzhou Huadu District Center for Disease Control and Prevention, Guangzhou 510080, China; 3Department of Clinical Molecular Biology, University of Oslo and Akershus University Hospital, 1478 Lørenskog, Norway; 4Radiology Center, Translational Medicine Center, Guangzhou Key Laboratory for Research and Development of Nano-Biomedical Technology for Diagnosis and Therapy, Guangdong Provincial Education Department, Key Laboratory of Nano-Immunoregulation Tumour Microenvironment, Central Laboratory, the Second Affiliated Hospital, Guangzhou Medical University, Guangzhou 510260, China

**Keywords:** nicotinamide adenine dinucleotide (NAD), nicotinamide riboside (NR), intestinal barrier, tight junctions (TJs), SirT1, mitochondrial function

## Abstract

Background: The epithelial tight junction is an important intestinal barrier whose disruption can lead to the release of harmful intestinal substances into the circulation and cause damage to systemic injury. The maintenance of intestinal epithelial tight junctions is closely related to energy homeostasis and mitochondrial function. Nicotinamide riboside (NR) is a NAD booster that can enhance mitochondrial biogenesis in liver. However, whether NR can prevent ethanol-induced intestinal barrier dysfunction and the underlying mechanisms remain unclear. Methods: We applied the mouse NIAAA model (chronic plus binge ethanol feeding) and Caco-2 cells to explore the effects of NR on ethanol-induced intestinal barrier dysfunction and the underlying mechanisms. NAD homeostasis and mitochondrial function were measured. In addition, knockdown of SirT1 in Caco-2 cells was further applied to explore the role of SirT1 in the protection of NR. Results: We found that ethanol increased intestinal permeability, increased the release of LPS into the circulation and destroyed the intestinal epithelial barrier structure in mice. NR supplementation attenuated intestinal barrier injury. Both in vivo and in vitro experiments showed that NR attenuated ethanol-induced decreased intestinal tight junction protein expressions and maintained NAD homeostasis. In addition, NR supplementation activated SirT1 activity and increased deacetylation of PGC-1α, and reversed ethanol-induced mitochondrial dysfunction and mitochondrial biogenesis. These effects were diminished with the knockdown of SirT1 in Caco-2 cells. Conclusion: Boosting NAD by NR alleviates ethanol-induced intestinal epithelial barrier damage via protecting mitochondrial function in a SirT1-dependent manner.

## 1. Introduction

The harmful use of alcohol is a cause of more than 200 diseases and can damage multiple organs and systems, especially the liver, brain and intestines. In particular, the intestine, as the main organ of ethanol absorption, is susceptible to serious injury from excess drinking.

In recent years, increasing evidence has shown that ethanol causes damage to organs or systems by affecting gut microbiology, disrupting intestinal barrier function and releasing intestinal contents and metabolites into the circulation [1,2,3]. An intact intestinal barrier is necessary to prevent excessive translocation of intestinal contents and metabolites into the circulation. The permeability plays an important role in the body’s interaction with the external environment and gut microbiota [4,5]. Therefore, it is of great importance to explore the factors that can protect intestinal barrier function to prevent and treat ethanol-induced injuries.

An important mechanism of epithelial barrier dysfunction is the energy deficiency caused by mitochondrial injury in intestinal epithelial cells (IECs). The formation and maintenance of the intestinal epithelial tight junctions (TJs) is regulated by cellular energy homeostasis. Sufficient cellular energy can promote the formation of TJs, thus maintaining the integrity of the intestinal epithelial barrier [6,7]. In addition, injured mitochondria increased the production of reactive oxygen species inducing cell damage, while the removal of reactive oxygen species produced by damaged mitochondria can restore intestinal epithelial barrier function [8]. Studies have shown that ethanol intake causes mitochondrial damage in the muscle, liver and heart and decreases ATP production in the skeletal muscle cells, hepatocytes and cardiomyocytes [9,10,11,12,13]. However, there are very few studies on the role of mitochondria in ethanol-induced intestinal barrier injury. Therefore, the exploration on how ethanol affects mitochondrial homeostasis in the intestinal epithelium may provide important insights for the prevention and treatment of ethanol-induced intestinal barrier damage.

It was reported that Sirtuin1 (SirT1) is a regulator for mitochondrial homeostasis, which is a deacetylase that regulates mitochondrial biosynthesis by enhancing the activity of peroxisome proliferator-activated receptor γ coactivator-1α (PGC-1α), a mitochondrial biogenesis coactivator, through its deacetylation [14,15,16]. Additionally, SirT1 is an important nicotinamide adenine dinucleotide (NAD)-depleting enzyme whose activation is highly dependent on intracellular NAD levels [17]. NAD is an essential coenzyme of mitochondrial oxidative phosphorylation and energy metabolism, which plays a key role in regulating metabolism and energy balance [18]. The decline of nuclear NAD levels can lead to impaired mitochondrial homeostasis [19]. In contrast, increased NAD levels are able to activate mitochondrial metabolism in liver, brown adipose tissue (BAT) and muscle to enhance mitochondrial homeostasis [20]. Previous studies demonstrated that ethanol inhibited the expressions and activities of SirT1 by reducing NAD levels in the liver and pancreatic β-cells [21], which in turn leads to liver mitochondrial damage by inhibiting the SirT1/PGC-1α axis [12]. Increasing NAD levels boosted the expressions and activities SirT1 in the liver and muscle and improved mitochondrial function [12,22]. However, it is still uncertain whether ethanol-induced intestinal damage is associated with inhibition of expression and activity of SirT1 induced by decreased NAD levels.

Recently, research spotlight has been focusing on the use of the active ingredients in food to prevent and treat diseases. Nicotinamide ribose (NR) is found in milk [23], which is a precursor of cellular NAD. NR is converted to nicotinamide mononucleotide (NMN) via nicotinamide ribose kinase (NRK), and then to NAD via nicotinamide mononucleotide adenylyl transferase (NMNAT) [24]. Many studies have shown that NR supplementation can increase NAD levels in the liver, brown adipose tissue and muscle, and enhance mitochondrial biogenesis and oxidative metabolism [20]. Moreover, NR has also been reported to have protective effects on intestinal health. NR supplementation can reverse ethanol-induced gut microbiota disorder, revitalizing intestinal stem cells in aged mice to repair intestinal damage [25,26]. In this study, we sought to investigate whether NR prevents ethanol-induced intestinal epithelial barrier injury, and whether it exerts a protective effect by modulating the SirT1 signaling pathway and mitochondrial function in intestinal epithelial cells.

## 2. Methods

### 2.1. In Vivo Experiments

All animal experiments were approved by the Animal Care and Utilization Committee of Sun Yat-sen University (2012-0080). Male C57BL/6J mice (4 weeks old) were purchased from Guangdong Experimental Animal Centre (Guangzhou, China). Mice were housed in a constant temperature environment free of specific pathogens with a 12 h light and 12 h dark cycle per day. The 8-week-old mice were randomly divided into three groups (*n* = 5–8): control group (CTRL), ethanol group (EtOH) and ethanol plus NR supplementation group (EtOH + NR). Mice in EtOH group and EtOH + NR group were subjected to a Lieber–DeCarli ethanol liquid diet (5% (*v*/*v*) ethanol-containing diet) (TROPHIC Animal Feed High-tech Co., Ltd., Nantong, China) for 10 days followed by gavage of a single dose of ethanol (5 g/kg·bw) to construct the chronic and binge ethanol feeding model (the NIAAA model), while CTRL group accepted a control liquid diet as previously described [27]. NR (ssss, Irvine, CA, USA) was administered to mice by daily gavage at a dosage of 400 mg/kg·bw. Mice were anaesthetized with pentobarbital sodium and blood was collected from the orbital sinus. The intestinal tissues were collected after cervical dislocation.

### 2.2. In Vitro Experiments

Caco-2 cells (within 10 generations) were grown in Minimum Essential Medium (MEM) (HyClone, USA). The medium was supplemented with 20% fetal bovine serum (FBS) (Gibco, USA) and 1% penicillin–streptomycin (Gibco, Carlsbad, CA, USA) at 37 °C under a 95% air/5% CO_2_ atmosphere. Caco-2 cells were divided into four groups: CTRL group, NR group, EtOH group and EtOH + NR group. NR was administered at a dosage of 0.5 mmol/L and ethanol was added at a dosage of 100 mmol/L. The duration of ethanol or NR intervention was 48 h. 

SirT1-siRNA (Santa Cruz, CA, USA) or Scramble-siRNA (Santa Cruz, Dallas, TX, USA) was used to transfect Caco-2 cells for 24 h, and then the transfection medium was discarded and replaced with normal medium with or without NR in the presence of ethanol for 48 h. Ethanol and NR were added as described above.

### 2.3. Histological Analysis 

Intestinal tissues were stained with hematoxylin and eosin (H&E) as well as immunohistochemistry. Briefly, ileum tissues were fixed in 4% paraformaldehyde and 5 μm thick paraffin-embedded sections were prepared to stain for H&E and perform immunohistochemistry. Immunohistochemistry and H&E staining were performed as previously described [12,28]. Representative images were obtained using TissueFAXSPlus ST (Meyer, Houston, TX, USA).

### 2.4. Biochemical Analysis 

To analyze the intestinal permeability of mice, fluorescein isothiocyanate (FITC)-dextran (4 kDa; Chondrex, Woodinville, WA, USA) was administered by gavage (20 mL/kg · bw) 4 h before sacrifice. Blood samples were collected immediately in EDTA-coated tubes after the mice were sacrificed and then centrifuged to obtain plasma. FITC-dextran was quantified by a spectrophotometer with an excitation wavelength of 490 nm and an emission wavelength of 520 nm. In addition, some blood samples were collected in EDTA-free tubes and then centrifuged to obtain serum. Serum LPS concentrations were measured using an endotoxin detection horseshoe crab kit (Bioend, Xiamen, Fujian, China).

### 2.5. Measurements of Metabolites

Distal ileum tissues were rapidly frozen in liquid nitrogen, powdered and then homogenized in ice-cold 80% (*v*/*v*) methanol/H_2_O solution. For Caco-2 cells, samples were washed with ice-cold PBS and then homogenized in ice-cold 80% methanol/H_2_O solution. After centrifugation, the vacuum dried samples were resuspended in acetonitrile/water (*v*/*v*) solution and analyzed by LC/MS.

Levels of NAD, ATP, ADP and AMP in ileum tissues and Caco-2 cells were determined using a UPLC-QTOF System (Infinity/6538, Agilent Technologies, Palo Alto, CA, USA) with a Hypercarb column (100 × 2.1 mm, 3 μm, Thermo Fisher Scientific, USA), as previously described [12,29]. 

### 2.6. RNA Isolation and Quantitative Real-Time PCR

Total RNA was extracted from ileum and Caco-2 cell by Trizol (Invitrogen, Carlsbad, CA, USA). The concentration and purity of total RNA were measured and approximately 1 μg RNA was reverse-transcribed into cDNA-by-cDNA Synthesis kit (TaKaRa, Otsu, Shiga, Japan). Quantitative real-time PCR was performed using SYBR Green Supermix kit (TaKaRa, Otsu, Shiga, Japan). Primers used in the study are listed in Table 1.

### 2.7. Analysis of Mitochondrial DNA (mtDNA) Content

mtDNA copy number was quantified by quantitative real-time polymerase chain reaction (qRT-PCR) as described previously [12]. Genomic DNA (gDNA) was isolated from ileum tissues and Caco-2 cells using the TIANamp Genomic DNA Kit (Tiangen, China) according to the manufacturer’s instructions. Relative content of mtDNA was determined by RT-qPCR. The ratio of mtDNA to nuclear DNA (nDNA) reflects the content of the mitochondria. nDNA was used as an internal standard as well as an indicator of nuclear genome content. Two mtDNAs (MT-CO1, ATPase6) and one nDNA (β-globin) were selected to quantify single-cell mtDNA copy number in Caco-2 cells and one mtDNA (ATPase6) and one nDNA (β-actin) were selected to quantify single-cell mtDNA copy number in mice ileum. Primers used in the study are listed in Table 1. 

### 2.8. Mitochondrial Function Assay

JC-1 dye was used to detect the cell membrane potential and MitoTracker red dye to detect biologically active mitochondria within the cells. After intervention with ethanol and NR for 48 h, Caco-2 cells were washed with PBS and then stained with JC-1 dye (Beyotime, Shanghai, China) or MitoTracker® Red CMXRos (Thermo Fisher Scientific, Carlsbad, CA, USA) according to the instructions. The staining was immediately followed by observation using a confocal microscope (Leica TCS SP5 Ⅱ).

### 2.9. Immunoprecipitation and Western Blot Analysis

Immunoprecipitation and western blot analysis were performed as previously described [30]. Occludin, SirT1, CS, SDHA, β-actin and acetylated lysine antibodies were from Cell Signalling; PGC-1α antibodies were from Novus Biologicals; ZO-1 and GAPDH antibodies were from Abcam; HRP-conjugated anti-rabbit and anti-mouse secondary antibodies were obtained from Jackson ImmunoResearch Inc. The blots were visualized using an enhanced chemiluminescence detection system according to the manufacturer’s instructions (ECL, Thermo Fisher Scientific, Carlsbad, CA, USA).

### 2.10. Statistical Analysis

Statistical analysis and graphing were performed using GraphPad Prism 8.3.0 (GraphPad Software, San Diego, CA, USA) and R v3.4.4 (R Foundation for Statistical Computing, Vienna, Austria). Statistical differences in quantitative data between groups were assessed by one-way ANOVA. Data were expressed as means ± SEM. *p* < 0.05 was considered as statistically significant. 

## 3. Results

### 3.1. NR Supplementation Alleviates Ethanol-Induced Intestinal Barrier Dysfunctions Both in In Vivo and In Vitro Experiments

We applied the chronic and binge ethanol feeding (NIAAA) model to test whether ethanol induced intestinal dysfunctions in mice. The body weight and food intake were similar among the three groups (Appendix A).

Intestinal permeability was evaluated by the levels of serum LPS and the fluorescein isothiocyanate (FITC)-dextran. As shown in Figure 1A, the LPS level in serum was elevated in EtOH group compared to the control (*p* < 0.01), while the LPS level in EtOH + NR group was lower than that in EtOH group (*p* < 0.01). The results of FITC-dextran assay showed that the concentration of FITC-dextran in the plasma was elevated in the ethanol-fed mice compared to that in the control mice, while NR supplementation suppressed the ethanol-induced increase of FITC-dextran in plasma (Figure 1B), indicating NR maintained the intestinal integrity in mice treated with ethanol. Considering that TJs are the major components in maintaining the physical barrier of the intestine, we examined the expressions of markers of TJs including ZO-1 and occludin in the ileum of mice by RT-qPCR and immunohistochemistry. Ethanol decreased the mRNA and protein levels of ZO-1 and occludin as compared with the control (*p* < 0.05), while the expressions of these two markers in the EtOH + NR group were similar to those of control (Figure 1C,D). In addition, the histopathological analysis of ileum H&E staining showed that the structure of the villi in the intestine was incomplete, and the structure of intestinal epithelial cells was destroyed in the EtOH group. In comparison, both the structures of the villi and the epithelial cells were intact in the NR supplement group, which were similar to those in the control group (Figure 1E). Taken together, above results demonstrated that NR treatment could mitigate intestinal barrier dysfunction induced by ethanol.

To examine whether NR has the potential to alleviate ethanol-induced disruption of epithelial barrier function in vitro, Caco-2 cells were used to investigate the effects of NR on ethanol-induced injuries. There were significant reductions in the expressions of ZO-1 and occludin in the Caco-2 monolayer after 48 h of exposure to 100 mM of ethanol compared with the control group (*p* < 0.05). However, with cocultivation of Caco-2 cells with 0.5 mM of NR for 48 hrs, ethanol did not decrease the expressions of ZO-1 and Occludin (*p* < 0.05) (Figure 1F). In addition, a decreased fluorescence intensity of the EtOH group was observed as compared with that of control (indicating decreased expressions of ZO-1 and occludin), while the fluorescence intensity of the EtOH + NR group was as strong as that of control (Figure 1G). Collectively, our data showed that NR supplementation alleviated ethanol-induced intestinal epithelial barrier damages both in in vivo and in vitro experiments.

### 3.2. NR Prevents Ethanol Induced Imbalance of NAD and Energy

Recent studies showed that the formation of TJs in IECs could be regulated by cellular energy homeostasis [31,32]. NAD is an important coenzyme of the tricarboxylic acid (TCA) cycle and is involved in the synthesis of energy. To figure out the molecular mechanism of NR supplement alleviating ethanol-induced intestinal barrier dysfunctions, we analyzed the NAD levels in distal ileum and Caco-2 cells. We found that ethanol treatment reduced NAD levels in both distal ileum and Caco-2 cells compared to that of control (*p* < 0.05). The NAD levels were similar in EtOH + NR group compared to the control group (Figure 2A,E). These findings indicated NR supplementation maintained the NAD levels in both the distal ileum and the epithelial cells, which was decreased by ethanol. Furthermore, our data showed that ethanol treatment reduced ileal ATP levels, accompanied with an increase in AMP levels and AMP/ATP ratio in contrast to the control group (*p* < 0.01) (Figure 2B–D). NR supplementation prevented the ethanol-induced disturbance of levels of ATP, AMP and the AMP/ATP ratio in the ileum (*p* < 0.05) (Figure 2B–D), indicating that NR efficiently alleviated ethanol-induced energy imbalance in the ileum. Similar results were found in Caco-2 cells that the levels of ATP and ADP and the ATP/ADP ratio were lower in ethanol-treated Caco-2 cells than in that of the control (*p* < 0.01). The levels of these indicators of energy homeostasis were similar in the EtOH + NR-treated cells compared to control (Figure 2F–H). Aforementioned results showed that NR played a role in regulating NAD and energy imbalance in intestinal epithelium induced by ethanol intake.

### 3.3. NR Restores Ethanol-Induced Intestinal Mitochondrial Dysfunctions Both in In Vivo and In Vitro Experiments

Mitochondria are the major organelles for energy production. We thus continued to test whether NR supplementation could regulate mitochondrial function in IECs. As we expected, significant reductions in protein expressions of succinate dehydrogenase (SDH) and citrate synthase (CS), two functional mitochondrial enzymes, were observed in ethanol-treated Caco-2 cells in comparison to the CTRL group. Note that these reductions were prevented by NR treatment (*p* < 0.05) (Figure 3A). JC-1 staining was used to detect changes in mitochondrial membrane potential and MitoTracker Red staining was used to specifically label biologically active mitochondria in Caco-2 cells. JC-1 aggregates and monomers were as indicated in red and green fluorescence, respectively. We found that ethanol-treated Caco-2 cells displayed weaker red fluorescence and brighter green fluorescence than that in control, indicating that mitochondria underwent depolarization by ethanol treatment. NR intervention reversed this switch, as evidenced by the increase in the red/green fluorescence intensity ratio compared to that of EtOH group (Figure 3B), indicating that NR prevented ethanol-induced decrease in mitochondrial membrane potential. Consistently, MitoTracker Red staining showed that ethanol induced a significant decline in the red fluorescence intensity (this represents the number of biologically active mitochondria) compared with the control, but NR upregulated the decrease in the red fluorescence intensity in ethanol-induced Caco-2 cells (Figure 3C). The above results suggest that ethanol-impaired intestinal mitochondrial function and NR supplementation prevented ethanol-induced mitochondrial damage. Further analysis revealed that ethanol decreased mitochondrial DNA (mtDNA) copy number compared with the CTRL group (*p* < 0.05), whereas NR supplementation prevented the ethanol-induced decreases in mtDNA copy number both in vivo and in vitro (*p* < 0.01) (Figure 3D,E), indicating that increased mitochondrial biogenesis may underpin the protective effect of NR on mitochondrial function. 

SirT1 is a deacetylase and promotes mitochondrial biogenesis through activation and deacetylation of PGC-1α, a major transcriptional coactivator regulating mitochondrial biosynthesis [33,34]. In vivo, the EtOH group showed significantly decreased mRNA levels of *SirT1* and *PGC-1α* in ileum of mice than that of control (*p* < 0.05), and these levels were significantly more elevated in the EtOH + NR group than the EtOH group (*p* < 0.01) (Figure 3F). In vitro, the protein expression of SirT1 was decreased after ethanol incubation compared with control, while NR supplementation suppressed this decline induced by ethanol (*p* < 0.05) (Figure 3H). The acetylation of PGC-1α was significantly elevated by ethanol treatment in Caco-2 cells. In contrast, the levels of acetylated PGC-1α were decreased in the EtOH + NR group compared to the EtOH group (*p* < 0.05) (Figure 3H). The above results demonstrated that both SirT1 activity and expression were lowered by ethanol but maintained by NR treatment. Correspondingly, both in vivo and in vitro, mRNA levels of *ERRα*, *NRF1*, *NRF2* and *TFAM* (these genes serve to regulate the downstream effects of PGC-1α on mitochondrial biosynthesis and energy metabolism) were reduced in the EtOH group. NR supplementation increased the levels of *ERRα*, *NRF1*, *NRF2* and *TFAM* compared to that in EtOH group (*p* < 0.05) (Figure 3G,I), suggesting mitochondrial biogenesis pathway was promoted by NR. NR treatment alone had no side effects on mitochondria of Caco-2 cells in this study, as the mtDNA copy number, the levels of PGC-1α and SirT1, mitochondrial membrane potential and the number of biologically active mitochondria of Caco-2 cells with NR intervention were similar to those of control (Figure 3A–C,E,H,I). The above data suggests that NR supplementation was capable of attenuating ethanol-induced intestinal mitochondrial dysfunctions and increasing mitochondrial biogenesis both in in vivo and in vitro experiments.

### 3.4. NR Improves Mitochondrial Function and Intestinal Barrier Injuries in a SirT1-Dependent Manner

To clarify the role of SirT1 on the modification of intestinal barrier function and mitochondrial function induced by ethanol, we applied *SirT1* siRNA to suppress the expression of SirT1 in Caco-2 cells; scrambled siRNA was applied in the control group. Caco-2 cells were further treated with ethanol, with or without NR treatment. *Sirt1* knockdown (*SirT1* KD) was confirmed by reduced protein expressions and mRNA levels of Sirt1 compared to cells transfected with scramble-siRNA (*p* < 0.01) (Figure 4A,B). In scramble-siRNA transfected cells, the levels of the acetylation of PGC-1α in NR+EtOH group were lower than that in EtOH group (*p* < 0.05), whereas in *SirT1*-KD cells, NR intervention failed to reduce ethanol-induced elevated level of the acetylated PGC-1α (Figure 4B), suggesting that NR promotes intestinal mitochondrial biosynthesis in a SirT1-dependent manner. Further assays of mitochondrial function showed that with *SirT1* knockdown, NR treatment did not increase the expressions of SDH and CS (Figure 4C). In addition, the mitochondrial function indicated by JC-1 staining further demonstrated that with *Sirt1* knockdown, NR lost its ability to restore the mitochondrial membrane potential (Figure 4D). Consistently, MitoTracker Red staining showed that in the presence of *SirT1* siRNA, the red fluorescence intensity was as weak in the NR group as the EtOH group, this differs from the obvious increase of fluorescence intensity in the NR group compared to the EtOH group in the presence of scramble siRNA (Figure 4E). Above results indicate that knockdown of SirT1 blocked the ability of NR to repair ethanol-induced mitochondrial damage. Next, we continued to test whether Sirt1 knockdown could block the protection of NR on TJs. As expected, in *SirT1*-KD cells, NR did not increase the expressions of both ZO-1 and occludin as in scramble-siRNA transfected cells (Figure 4F,G). Taken together, these data indicate that SirT1 played an important role in the protective effects of NR against both mitochondrial dysfunctions and intestinal barrier injuries induced by ethanol.

## 4. Discussion

Intestinal dysfunction including gut dysbiosis and barrier disruption contributes to the development of diseases in the liver and other organs. The intestinal barrier consists of physical, secretory, immunological and microbic components. TJs are the important components of epithelial barrier, whose integrity is essential in blocking gut microbes and adverse products such as LPS translocated to the circulation. In this study, we focused on the intestinal barrier function affected by ethanol and NR. Our study reports for the first time that ethanol induces intestinal epithelial barrier injury via destroying mitochondrial function. Supplementation of NR, which is a NAD precursor, protects against ethanol-induced intestinal epithelial injuries via maintaining mitochondrial function by promoting mitochondrial biogenesis in a SirT1-dependent manner. 

According to our previous report, NR as a NAD supplement was effective in preventing ethanol-induced liver injury. We found that NR has direct protective effect on the liver. Furthermore, we detected that NR diminished the ethanol-induced release of LPS into the circulation, suggesting NR also has the potential to protect intestinal function. Our current report confirms the hypothesis that NR can protect intestinal barrier function, which widens the view of the protective mechanisms of NR. 

Increased intestinal permeability allows pathogenic bacteria and microbial-associated metabolites (e.g., LPS) to enter circulation from the intestine. According to a previous study, rats of daily ethanol gavage show elevated levels of urinary lactose as early as week 2, indicating increased intestinal permeability and increased serum LPS levels at week 4, leading to endotoxemia [1]. In our experiment, we also found that mice fed an ethanol diet showed reduced TJ protein expressions and damaged intestinal epithelial villi, resulting in increased intestinal permeability accompanied by an increase in serum LPS level. NAD supplementation by NR can effectively reduce ethanol-induced intestinal hyperpermeability and prevent LPS release into the blood. As a metabolic cofactor, NAD is the rate-limiting co-substrate of many enzymes in various biological processes such as energy metabolism, stress adaptation and cell homeostasis. Several recent articles showed that NAD plays a key role in the maintenance of intestinal function. Depletion of NAD in intestine of cisplatin-treated mice was increased, and the use of dunnione as a strong substrate for quinone oxidoreductase 1 increased intracellular NAD levels, which prevented cisplatin-induced structure damage of intestinal villi [35]. Increased NAD levels by NaMN, NaAD and NAD in the LS 174 T human goblet cell line promoted the secretion of MUC2, which protected the mucus barrier of the intestine [36]. However, a decrease in intestinal NAD levels caused by ethanol ingestion has not been reported. Notably, we report for the first time that ethanol consumption leads to a decrease in ileal NAD levels and that boosting ileal NAD levels by NR supplementation alleviates ethanol-induced epithelial barrier damage. Our data widen the view of the protective effects of boosting of NAD on intestinal functions.

The maintenance of the intestinal epithelial barrier requires large amounts of energy [31]. Intestinal epithelial cells consume up to 20% of total ATP in barrier maintenance [32]. It is known that mitochondria are the main organelles that provide energy needed by most cells, and its impaired function leads to deficient energy to maintain the intestinal epithelial barrier. Previously, Chauhan et al. found that the intestines of long-term ethanol-fed rats showed impairment of mitochondria [37]. We found that short-term ethanol consumption can also damage intestinal mitochondria, and the mechanism may be related to the inhibition of intestinal mitochondrial biosynthesis. PGC-1α is generally regarded as the main regulator of mitochondrial biosynthesis [38,39]. Although PGC-1α has been reported in the literature to be abundantly expressed in IECs and to play an important role in protecting intestinal barrier function [40], the function of PGC-1α has not been reported in models of ethanol damage. In the present study, ethanol reduced intestinal mRNA levels of PGC-1α and the levels of deacetylation of PGC-1α in Caco-2 cells. NAD supplementation promoted PGC-1α deacetylation by activating intestinal SirT1 expression and its deacetylation activity, thereby regulating mitochondrial biosynthesis in intestinal epithelial barrier. SirT1 is widely expressed in IECs of the small intestine and colon. SirT1 is a NAD-dependent enzyme, and its activation is highly dependent on NAD levels, suggesting the modulation of NAD is emerging as a valuable tool to regulate SirT1 function. As we have found so far, NAD supplementation by NR prevents ethanol-induced decrease in intestinal SirT1 levels. SirT1 is a key deacetylation enzyme in cells that regulates intestinal damage by modulating the acetylation status of multiple transcriptional regulators such as P65, Foxp3α and PGC-1α [41,42,43]. Increasing evidence from human and rodent studies suggests that colitis, microbiota dysbiosis and intestinal barrier damage are closely associated with impairment of intestinal SirT1 signaling pathway [44,45,46]. Activation of SirT1 in vivo and in vitro can alleviate impairment of intestinal function. Intervention with SirT1 activators such as Cay10591 or SRT1720 has been reported to be able to activate intestinal SirT1 to prevent and cure experimental colitis [47,48,49]. In addition, resveratrol and its derivative pterostilbene, activators of SirT1, has been found to ameliorate oxidative stress-induced intestinal mitochondrial damage [41]. In our study, we found that NR enhanced intestinal SirT1 expression and its deacetylation activity, which subsequently regulate mitochondrial function and protected intestinal epithelial barrier function damaged by ethanol intake. However, the ability of NR to ameliorate ethanol-induced mitochondrial and intestinal barrier damage was lost after SirT1 knockdown. Furthermore, similar to our findings that in aged mice models, NR revives intestinal stem cells and enhances their ability to repair intestinal damage, but the protective effects of NR are blocked by the SirT1 inhibitor EX527 [25]. These suggest the alleviation of intestinal injury by NR is via SirT1 signaling pathway to a considerable extent. Given the fact that at the dosage of NR used in our experiments, no adverse effects were observed in previous clinical trials or animal experiments [26,50,51], a separate NR supplementation group was thus not established in the in vivo experiments of this study. Our study also has a limitation. Given that female rodents are more susceptible to ethanol-induced liver injury than male rodents, to avoid death of mice during ethanol feeding, we chose male mice to perform experiments. Therefore, we did not compare the sex differences in NR-protective effects on ethanol-induced intestinal barrier damage.

In brief, our data indicate that boosting NAD by NR prevents ethanol-induced intestinal dysfunction by activation of epithelial mitochondrial function. The protective effects of NAD on intestinal barrier depend on the activation of SirT1 to a considerable extent. NR acts as a good NAD precursor without side effects and is able to regulate mitochondrial function via activating the SirT1 signaling pathway, maintain TJs stability and ultimately reverse ethanol-induced damage to the intestinal epithelial barrier.

## Figures and Tables

**Figure 1 nutrients-15-00174-f001:**
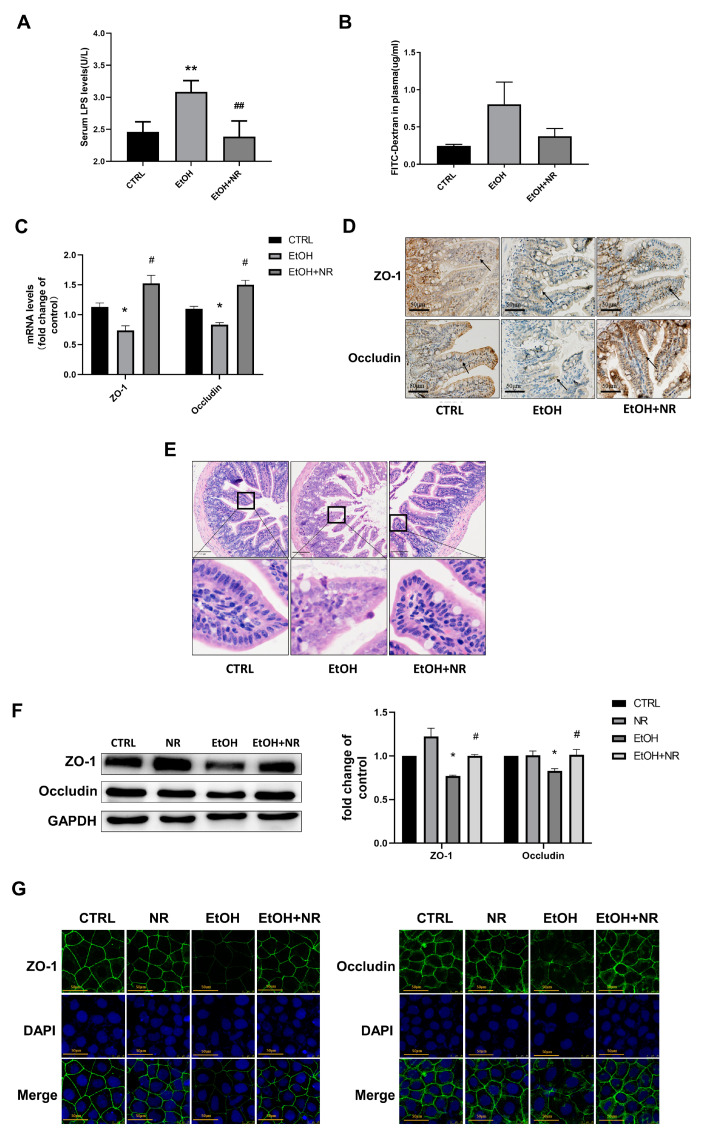
NR supplementation alleviates ethanol-induced intestinal barrier dysfunctions both in in vivo and in vitro experiments. (**A**) The Serum LPS levels after sacrifice; (**B**) the plasma FITC-Dextran concentrations after sacrifice of the assessment of in vivo gut permeability; (**C**) relative mRNA levels of ZO-1 and occludin related to TJs in ileum determined by qRT-PCR. Results were expressed as fold changes of CTRL; (**D**) representative images of immunohistochemistry of histological ileum sections for ZO-1 and occludin with 200× magnification; (**E**) representative images of histological ileum sections after H&E staining with 100× magnification. Mice in EtOH group and EtOH + NR group were fed a Lieber–DeCarli ethanol diet for 10 d followed by gavage of a single dose of ethanol with or without NR treatment. Mice in the CTRL group were fed control liquid diet. Data are shown as mean ± SEM (*n* = 5–8/group), * *p* < 0.05 and ** *p* < 0.01 compared with the CTRL group; # *p* < 0.05 and ## *p* < 0.01 compared with the EtOH group. (**F**) The representative images and quantification of Western blot analysis of ZO-1 and occludin protein expressions related to TJs in Caco-2 cells. GAPDH was used as an internal control. Results were expressed as fold changes of CTRL. (**G**) Representative images of immunofluorescence of ZO-1 and occludin related to TJs in Caco-2 cells with 1200× magnification. Caco-2 cells were treated with or without 0.5 mmol/L NR in the presence or absence of 100 mmol/L ethanol for 48 hrs. Data are presented as mean ± SEM (*n* = 4/group). * *p* < 0.05 and ** *p* < 0.01 compared with the CTRL group; # *p* < 0.05 and ## *p* < 0.01 compared with the EtOH group.

**Figure 2 nutrients-15-00174-f002:**
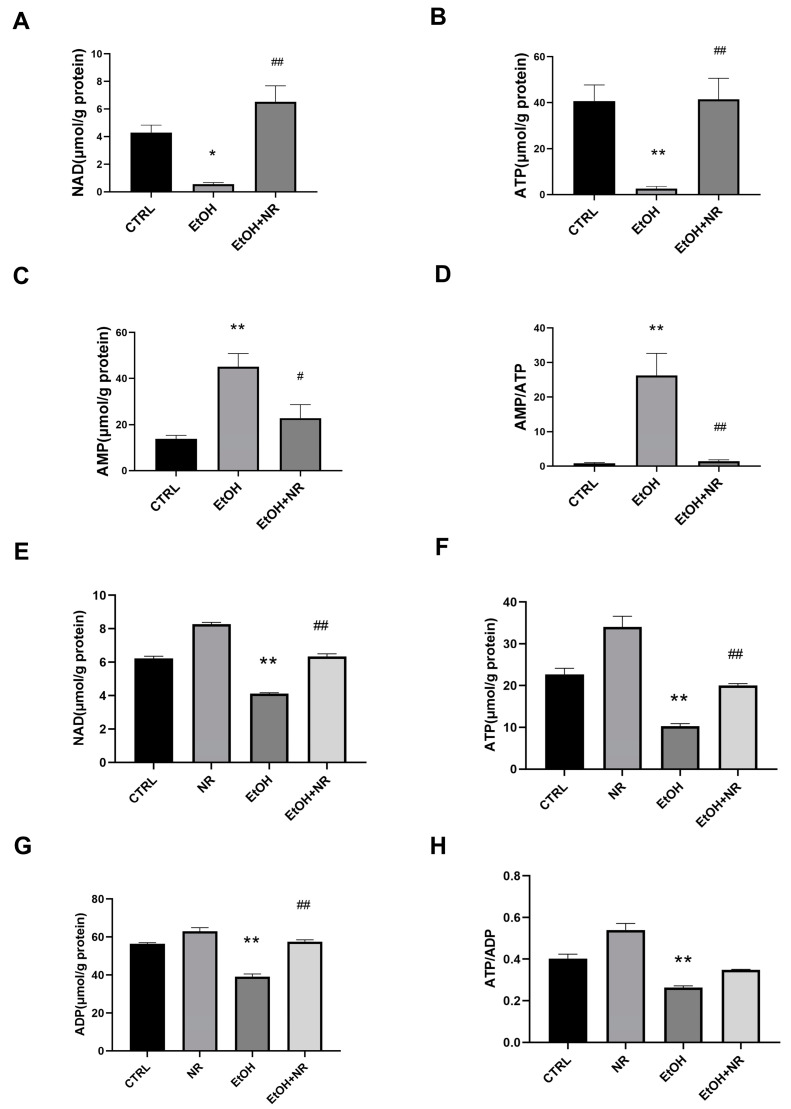
NR prevents ethanol-induced NAD and energy imbalance both in in vivo and in vitro experiments. (**A**) The LC/MS metabolomic analysis of NAD levels of terminal ileum. (**B**) The LC/MS metabolomic analysis of ATP levels of terminal ileum. (**C**) The LC/MS metabolomic analysis of AMP levels of terminal ileum. (**D**) The LC/MS metabolomic analysis of AMP/ATP ratios of terminal ileum. Mice in EtOH group and EtOH + NR group were fed a Lieber–DeCarli ethanol diet for 10 d followed by gavage of a single dose of ethanol with or without NR treatment. Mice in the control group were fed control liquid diet. Data are presented as mean ± SEM. n = 5–8/group, * *p* < 0.05 and ** *p* < 0.01 compared with the CTRL group; # *p* < 0.05 and ## *p* < 0.01 compared with the EtOH group. (**E**) The LC/MS metabolomic analysis of NAD levels in Caco-2 cells; (**F**) the LC/MS metabolomic analysis of ATP levels in Caco-2 cells; (**G**) the LC/MS metabolomic analysis of ADP levels in Caco-2 cells; (**H**) the LC/MS metabolomic analysis of ATP/ADP ratios in Caco-2 cells. Caco-2 cells were treated with or without 0.5 mmol/L NR in the presence or absence of 100 mmol/L ethanol for 48 hrs. Data are presented as mean ± SEM (n = 4/group). * *p* < 0.05 and ** *p* < 0.01 compared with the CTRL group; # *p* < 0.05 and ## *p* < 0.01 compared with the EtOH group.

**Figure 3 nutrients-15-00174-f003:**
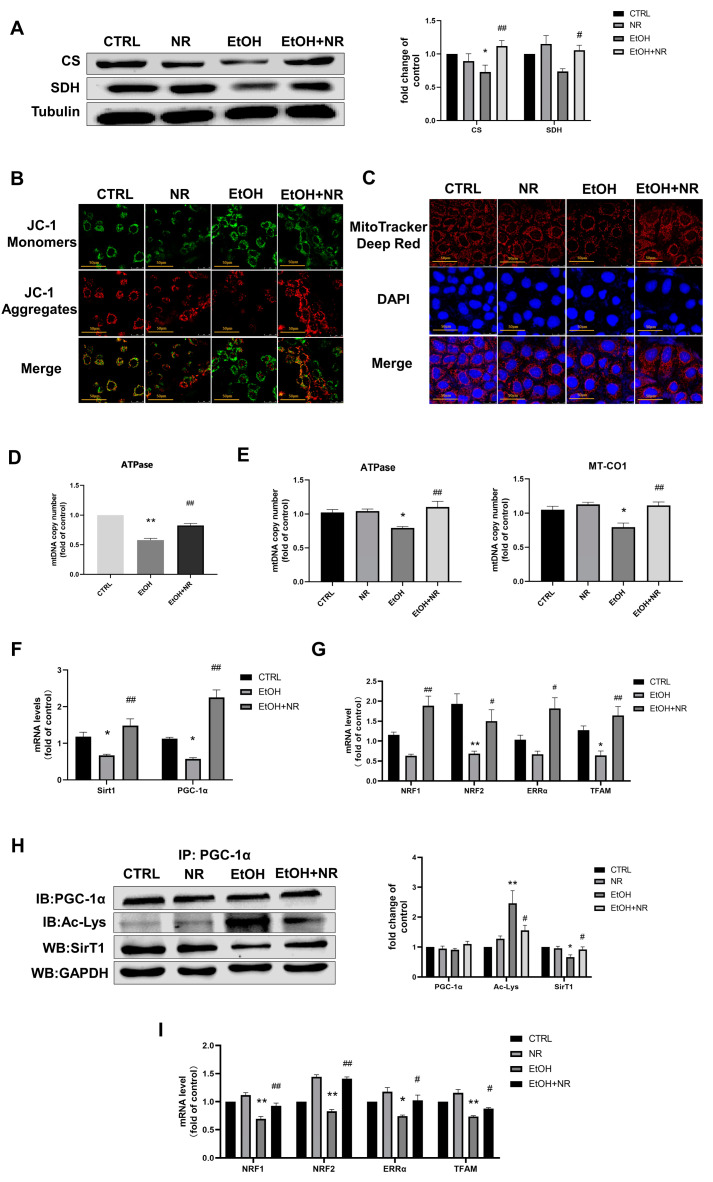
NR restores ethanol-induced intestinal mitochondrial dysfunctions both in in vivo and in vitro experiments. (**A**) The representative image and quantification of Western blot analysis of CS and SDH protein expressions in Caco-2 cells. Tubulin was used as an internal control. Results are expressed as fold changes of CTRL. (**B**) Representative images of JC-1 staining with 1200× magnification of Caco-2 cells. Green fluorescence: JC-1 monomers; red fluorescence: JC-1 aggregates. (**C**) Representative images of MitoTracker Deep Red staining with 1200× magnification of Caco-2 cells. MitoTracker Deep Red: mitochondria; DAPI: nucleus. (**D**) Mitochondrial DNA (mtDNA) copy number in the ileum determined by qRT-PCR. mtDNA: ATPase; nDNA: β-actin. Results are expressed as fold changes of CTRL. (**E**) Mitochondrial DNA (mtDNA) copy number in Caco-2 cells determined by qRT-PCR. mtDNA: ATPase and MT-CO1; nDNA: β-globin. Results are expressed as fold changes of CTRL. (**F**) Relative mRNA levels of SirT1 and PGC-1α in the ileum determined by qRT-PCR. Results are expressed as fold changes of CTRL. (**G**) Relative mRNA levels of NRF1, NRF2, ERRα and TFAM in the ileum determined by qRT-PCR. Results are expressed as fold changes of CTRL. (**H**) The representative image and quantification of immunoprecipitation and Western blot analysis of SirT1, total PGC-1α and acetylated PGC-1α protein expressions in Caco-2 cells. GAPDH was used as an internal control. Results are expressed as fold changes of CTRL. (I) Relative mRNA levels of NRF1, NRF2, ERRα and TFAM mRNA levels in Caco-2 cells determined by qRT-PCR. Results are expressed as fold changes of CTRL. Mice in EtOH group and EtOH + NR group were fed a Lieber–DeCarli ethanol diet for 10 d followed by gavage of a single dose of ethanol with or without NR treatment. Mice in the CTRL group were fed control liquid diet. Data are presented as mean ± SEM (*n* = 5–8/group). Caco-2 cells were treated with or without 0.5 mmol/L NR in the presence or absence of 100 mmol/L ethanol for 48 hrs. Data are presented as mean ± SEM (*n* = 4/group). * *p* < 0.05 and ** *p* < 0.01 compared with the CTRL group; # *p* < 0.05 and ## *p* < 0.01 compared with the EtOH group.

**Figure 4 nutrients-15-00174-f004:**
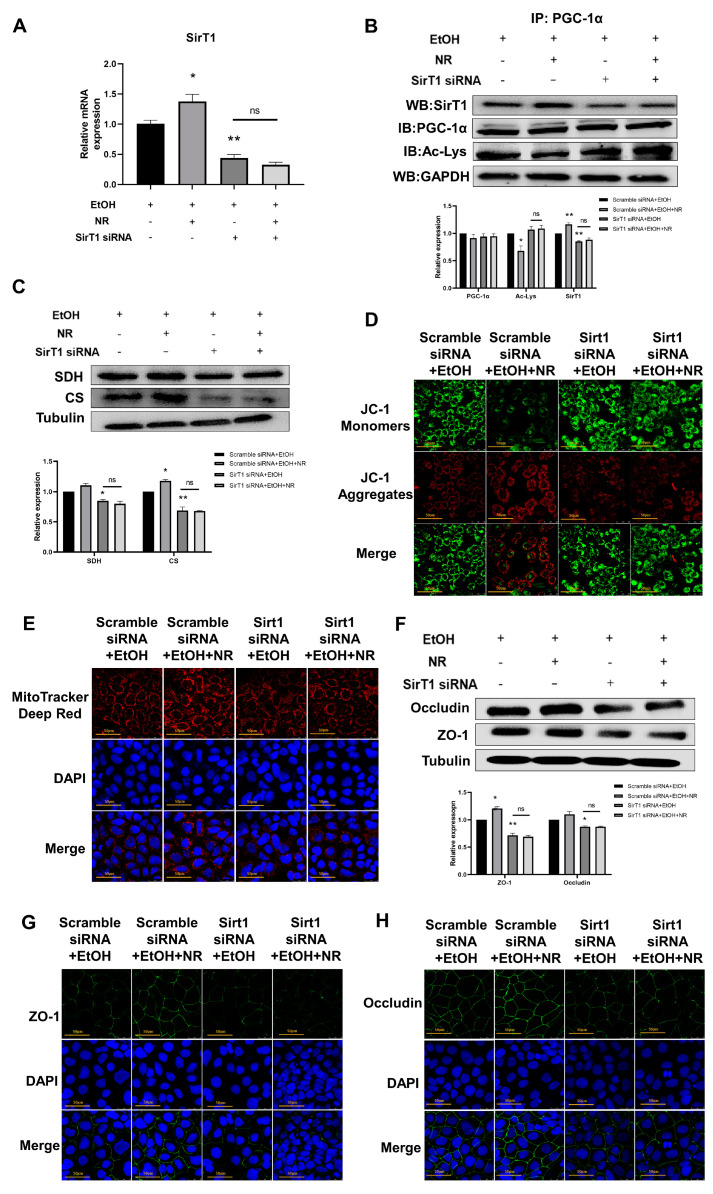
NR improves mitochondrial function and intestinal barrier injury in a SirT1-dependent manner. (**A**) Relative mRNA levels of SirT1 mRNA expression in Caco2 cells determined by qRT-PCR; results were expressed as fold changes of scramble siRNA-transfected cells without NR. (**B**) The representative image and quantification of immunoprecipitation and Western blot analysis of SirT1, total PGC1α and acetylated PGC-1α protein expressions in Caco-2 cells. GAPDH was used as an internal control. (**C**) The representative image and quantification of Western blot analysis of CS and SDH protein expressions in Caco-2 cells. Tubulin was used as an internal control. (**D**) Representative images of JC-1 staining with 1200× magnification of Caco-2 cells. Green fluorescence: JC-1 monomers; red fluorescence: JC-1 aggregates. (**E**) Representative images of MitoTracker Deep Red staining with 120× magnification of Caco-2 cells. MitoTracker Deep Red: mitochondria; DAPI: nucleus. (**F**) Representative images and quantification of Western blot analysis of ZO-1 and occludin protein expressions related to TJs in Caco-2 cells. Tubulin was used as an internal control. (**G**) Representative images of immunofluorescence of ZO-1 and occludin related to TJs in Caco-2 cells with 1200× magnification. (H) Representative images of immunofluorescence of occludin related to TJs in Caco-2 cells with 1200× magnification. Caco-2 cells were transfected with SirT1 siRNA or scramble siRNA for 24 h and then the transfection medium was discarded and replaced with normal medium with or without 0.5 mmol/L NR in the presence of ethanol added at a dose of 100 mmol/L for 48 h. Data are presented as mean ± SEM (*n* = 3/group). * *p* < 0.05 and ** *p* < 0.01 compared with the Scramble siRNA + EtOH group. ns means there is no statistical difference between the two groups.

**Table 1 nutrients-15-00174-t001:** Quantitative real-time PCR primers.

Gene	Species	Forward Primer	Reverse Primer
SirT1	mouse	TGTGAAGTTACTGCAGGAGTGTAAA	GCATAGATACCGTCTCTTGATCTGAA
PGC-1α	mouse	AAGTGTGGAACTCTCTGGAACTG	GGGTTATCTTGGTTGGCTTTATG
ERRα	mouse	GGGGAGCATCGAGTACAGC	AGACGCACACCCTCCTTGA
NRF1	mouse	AGCACGGAGTGACCCAAAC	AGGATGTCCGAGTCATCATAAGA
NRF2	mouse	CTTTAGTCAGCGACAGAAGGAC	AGGCATCTTGTTTGGGAATGTG
TFAM	mouse	AACACCCAGATGCAAAACTTTCA	GACTTGGAGTTAGCTGCTCTTT
ATPase	mouse	GCCCGAGTGTATGGGACAG	GGACACACCAATAATGAGGTGC
β-actin	mouse	GTGGTGGTGAAGCTGTAGCC	AGCCATGTACGTAGCCATCC
SirT1	human	TCTGGCATGTCCCACTATCA	TAGGCGGCTTGATGGTAATC
PGC-1α	human	GCTTTCTGGGTGGACTCAAC	CTGCTAGCAAGTTTGCCTCA
ERRα	human	TCCAGCTCCCACTCGCTGCC	ACACTCGTTGGAGGCCGGAC
NRF1	human	CCAGTTTAGTGGGTGGTAGG	CGGGAGCTTTCAAGACATTC
NRF2	human	GGCGCGTAGGTTTGTTCTAC	ACTCCAGCCATGACTAAAAGAGA
TFAM	human	GTCACTGCCTCATCCACC	CCGCCCTATAAGCATCTT
ATPase6	human	TCAGCCTACTCATTCAAC	CTAGGATAGTCAGTAGAATTAGA
MT-CO1	human	CACACGAGCATATTTCAC	GTACGATGTCTAGTGATGA
β-Globin	human	AAAGGTGCCCTTGAGGTTGTC	TGAAGGCTCATGGCAAGAAA
β-actin	human	TCAAGAAAGGGTGTAACGCAACT	CGACAGGATGCAGAAGGAGAT

## Data Availability

The datasets generated during and/or analyzed during the current study are available from the corresponding author upon reasonable request.

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
