# Peer review of "NAD Supplement Alleviates Intestinal Barrier Injury Induced by Ethanol Via Protecting Epithelial Mitochondrial Function"

_nutrients, 2022, doi:10.3390/nu15010174_

Round 1

Reviewer 1 Report

Great job in illustrating the protective effect of NAD supplement against ethanol induced intestinal barrier injury.

I have critically reviewed this manuscript titled “NAD supplement alleviates intestinal barrier injury induced 2 by ethanol via protecting epithelial mitochondrial function” I have come to the following conclusions:

This manuscript has a good scientific merit and interpretation/discussion and correlation is appropriate. However, some general issues need to be addressed and sentence/and typing/technical errors need to be revised.

My first and foremost suggestion in this paper is that it is required to improve the abstract to understand the full manuscript.

The following sections also need minor corrections and questions as per following comments:

 Did you perform any female mice experiments to see the sex differences?  

If you do have the serum available from your experiment then please include the serum amyloid A one of the important parameters in this manuscript.

Why did author not check the gene expression in in vivo?

 How many biological replicates have been taken for real time pcr in this paper?

Could you please mention the passage number of caco-2 cell lines that author used for the experiment?

Please modify the figures as anyone can’t read the figure text words properly. Font size is so small. Improve the figure quality as well.

Explain the cell culture experiment methods section.

Rewrite the reference: Lopes, F., Keita, A. V., Saxena, A., Reyes, J. L., Mancini, N. L., & Al Rajabi, A., et al. (2018). ER- 544 stress mobilization of death-associated protein kinase-1-dependent xenophagy counteracts mitochon- 545 dria stress-induced epithelial barrier dysfunction. JOURNAL OF BIOLOGICAL CHEMISTRY, 546 293(9), 3073-3087.http://doi.10.1074/jbc.RA117.000809

it is not written in correct way.

Add some recent references.

Reviewer 2 Report

Study by Wenli Le et al., entitled "NAD supplement alleviates intestinal barrier injury induced by ethanol via protecting epithelial mitochondrial function" shows the protective effect of NAD supplementation in alleviating intestinal barrier injury and mitochondrial function. Study is very well conducted. However, there are some concerns that can be addressed to strengthen the article.

1) Fig.1 can be moved to a supplementary figure since it does not contribute to the data significantly.

2) Since EtOH is also known to induce inflammation, some inflammation markers and cytokines can be a good parameter if NAD is also reversing the inflammation. 
